# The What, Why and How of Child Participation—A Review of the Conceptualization of "Child Participation" in Child Welfare

**Berit Skauge \***, **Anita Skårstad Storhaug** and **Edgar Marthinsen**

Department of Social Work, Norwegian University of Science and Technology (NTNU), 7491 Trondheim, Norway; anita.s.storhaug@ntnu.no (A.S.S.); edgar.marthinsen@ntnu.no (E.M.)
\* Correspondence: berit.skauge@ntnu.no

**Abstract:** This review explores the conceptualization of "child participation" in a child welfare context. The analyses are based on the theories, models and concepts researchers apply when framing their studies. Central to the authors' conceptualizing is the understanding of why children should participate. Children's rights are a common starting point for many authors, but they differ on whether children should participate out of consideration for children's intrinsic value (e.g., concern for their well-being) or for the instrumental value of the participation itself (e.g., service outcome). The analysis also focuses on how authors measure participation level. The analysis showed that most authors presented a limited rights-focused goal for the collaboration with children, while a minority group problematized the concept. Although several researchers emphasize that participation requires a process, few authors see the meaning-making process as the main purpose of child participation.

**Keywords:** child participation; child welfare; conceptualization; citizenship; children's rights; meaning making

## 1. Introduction

In recent years, the concept of "participation" has become firmly embedded in discourse on children's rights, public policy and research across the world. Central in this regard is article 12 of the United Nations Convention on the Rights of the Child, which states that children have the right to participate in matters that have a significant impact on their lives. Although this increased focus on children's rights to participation has led to changes in legislation in individual countries, studies have shown that social workers find it challenging to implement children's participation in practice (Van Bijleveld et al. 2015; Kennan et al. 2019). Child welfare and child protection services (we use the term "child welfare", sometimes shortened to CW, throughout the article) is an arena in which child participation is central because of the often-invasive nature of interventions in the child's life.

In academic research, there has been an increased focus on children's participation in a CW context. However, the concept of participation seems to be regarded by many as lacking clarity and precision (Križ and Skivenes 2017; Landsdown 2010; Sinclair 2004) and as a concept that is often used without being theoretically and contextually explained and operationalized (Vis and Thomas 2009; Tingstad 2019). Social workers also have different understandings of the concept of participation (Vis et al. 2012), which may explain the gap between the general perception that children should always participate and actual practice. Van Bijleveld et al. (2015) claim that lack of clarity in what the concept entails among scholars might be a key barrier to child participation. Reynaert et al. (2009) and Quennerstedt et al. (2018) highlight the importance of a transparent and exploratory debate to counter the distortion and simplification of a complicated field.

According to Bal (2002), concepts travel between disciplines, individual scholars, time and space, and are neither simple nor adequate in themselves. If they are carefully considered, they can offer miniature theories, and can be useful analytical tools. Bal claims that we tend to use some overarching concepts as if their meanings were clear-cut and

common, and we tend to take for granted a certain use and understanding of concepts. This leads to confusion, which increases, according to Bal, with concepts that are close to ordinary language (2002, p. 21) We consider "participation" to be such a concept. If concepts are (mis)used as ordinary words or labels, they may lose their working force and become meaningless (Bal 2002, p. 19). Because concepts used in research publications and CW practice are a key to intersubjective understanding, they need to be explicit, clear and defined.

This review does not focus on the results of the included studies. Rather, the aim is to explore how the concept of "child participation" in a CW context is understood and used by authors, by critically assessing research conducted over the past 10 years. A central question is: What conceptualizations and operationalizations of children's participation do we find in the reviewed articles? We do not search for possible unambiguous definitions, but how the term is explored in terms of how it is operationalized, explained and understood. We aim to bring insight into different aspects and understandings of child participation that can contribute to an increased intersubjective understanding of the concept.

## 2. The Various Contexts of Child Welfare Services

This review includes articles from a diversity of countries and cultures. Various countries' CW systems differ when it comes to legislation, working methods and approaches. Western CW systems are mainly categorized in two types: risk-oriented or service-oriented. Risk-oriented systems have a high threshold for intervention and focus on mitigating serious risk to children's safety and health. The United States and England are examples of this approach. Service-oriented systems have a low threshold for interventions and aim to promote safe and healthy childhoods by prioritizing preventive in-home measures. Examples of this type of system can be found in Norway, Sweden and Finland (Falck-Eriksen and Skivenes 2019; Burns et al. 2017). Service-oriented systems are also often described as child-centered, as opposed to family-oriented. One feature of child-centered CW organizations is that children's rights, interests and needs are considered as essential elements in service-provision. Children are given the opportunity to express their views and opinions and to be involved in matters concerning them (Pösö and Enroos 2017). This systematization has been criticized for being limited to high-income countries and for not being sufficiently nuanced, and for inadequately acknowledging cultural complexity and different CW contexts (Connolly et al. 2014: Waldock 2016). The works of Connolly et al. (2014) and Connolly and Katz (2020) aim to develop a typology of child protection systems across a wide range of countries, including low- and middle-income countries. They argue that there are two central dimensions that are useful in explaining child protection systems development across international countries; individual or community focus and more or less regulated.

Another theme relevant to context is that, generally, the CW process seems to be divided into three chronological phases in the work process: the reception of reports of concern, the assessment phase, and the intervention phase (Pölkki et al. 2012; Heimer et al. 2017). The teams involved, the working process and the purpose for involving the child may differ in each phase of the CW process. Different national-cultural and organizational contexts may influence the opportunities and limitations of children's participation.

A key issue for this review concerns the authors' contextualization of participation. Beyond theoretical connection, the study focuses on whether they explain the studies' context when it comes to culture, phase of the child welfare process, characteristics of the child (such as age and gender), and the different contexts where CW services cooperate with children.

## 3. Methods

We have chosen a scoping approach to map the key concepts that underpin the research area. The review was conducted to identify, examine and clarify definitions and in the conceptualization of the term used in child welfare studies. We try to provide

an overview of how the complexity of the field of child participation in child welfare is described, including the different definitions and interpretations of the term. The aim is to determine the scope and coverage of the concept of child participation and give indication of the volume of literature and studies available. To capture the broader scope of the interpretations of child participation practices, a variety of studies are included.

This review was conducted through the following stages inspired by scoping views (Arksey and O'Malley 2005; Munn et al. 2018): formulating research questions, searching databases using identified keywords, assessing articles for relevance, screening according to inclusion and exclusion criteria, and encoding and analyzing the final sample. In line with Harder and Thomas (2010), the synthesis can also be described as critical, as it questions the way the authors of the included studies conceptualize and construct the concept of children's participation in CW.

### 3.1. Inclusion and Exclusion Criteria

The material of this review consists of articles reporting original empirical studies related to children's participation in a CW context, published in peer-reviewed journals, in English or a Scandinavian language between May 2009 and February 2020. Originally, studies that did not report on primary studies were an exclusion criterion. This was changed in order to gain a greater breadth of interpretations of the concept of child participation. The time frame was selected in order to review recent status of knowledge, and to limit the search results. Publications were excluded if they were a conference abstract, book, thesis, book chapter, or popular press article and if the full text was unavailable via our University library subscriptions.

### 3.2. Search Strategy

The first step was an initial search in a selection of relevant databases, followed by an analysis of the words in the title and summary, and of the keywords used to describe the articles. This is how we came to the keywords that we have used in further data searches.

Searches were conducted through the following electronic databases: Oria, Web of Science, Idunn, JStOR, Scopus, Web of Science, Google Scholar, and ScienceDirect.

Different combinations of the following keywords were used as search terms: Child participation, OR involving, OR consulting, OR engage, OR cooperation, OR collaboration, AND assessment, OR in-home interventions, OR measures, OR support, OR in care, AND child protection, OR child welfare. Reference lists of the included articles were reviewed to check for missing studies of relevance. The peer-reviewed articles identified in these searches were briefly examined (title and abstract) prior to download ($N = 1496$). Papers identified as potentially relevant for this review, after excluding duplicates and articles, clearly irrelevant to the field of child welfare services, were screened by the first author. In total, 318 articles remained after this. A more thorough examination of title and abstract, and sometimes the full text of these, conducted by the first and second author, also excluded articles that were irrelevant to the review, leaving us with 86 articles. Examples of articles excluded in this process are articles focusing on child participation in the justice and health care system; participation mainly with parents; development of communication skills with children and use of language interpreters in conversations with children

### 3.3. Data Extraction and Analysis

When it was unclear whether a study was eligible for inclusion based on the information presented in the title or abstract, an independent review was conducted from B.S. and A.S.S. ($N = 20$). A selection of articles was read by the second author, with the purpose of making sure that both authors had the same understanding of the inclusion criteria. These articles were then read more thoroughly, leaving a total of 44 articles. In total, 40 articles met the inclusion criteria. In total, 4 articles were identified from reference lists, and those that did not report on a primary study, but analyzed several studies, as well as articles that used empirical data from their own studies in a theoretical article, were included.

Disagreements (*N* = 3) were clarified through discussion of the rationale for each analysts' choice to include or exclude an article. The final collection of studies represents 13 countries and 5 continents.

The main findings related to our research questions were extracted, and the studies were categorized in the following way: (1) How is participation explained and conceptualized? Are the studies contextualized? (as it relates to cultural/national background, phase of the child welfare process, characteristics of the child, like age or gender), and the different contexts where CW services cooperate with children. (2) What indicators of participation are used? (3) What do the authors express about the purpose of participation? Since the review includes 44 articles, we do not provide a description of each included article, but our results from the analysis are illustrated through examples.

An overview of how child participation has been described was presented, including the varying definitions and interpretations of the term. We conducted an analysis of the conceptual features of extracted data. With regard to the conceptual analysis, we focused on examining common and unique themes among definitions of child participation and their operationalization.

### 3.4. Study Limitations

The study has potential limitations related to the question: Have we found all relevant publications for our research questions? Different selection criteria and search words could include other interesting studies. For example, we have limited our inclusion criteria to peer-reviewed empirical articles. Broader inclusion criteria in this regard could have included relevant reports, books and theoretical articles. A review that focuses on the concepts being used rather than reporting on the studies' findings may be more open to subjective interpretations from the authors conducting the review. Other researchers could also interpret the concepts in other ways than we do. We have tried to be transparent by explaining our systematic approach, and by being open about the interpretations we have made. The review includes many studies, which can be a strength as it provides a wide overview of this topic of research. At the same time, the large number implies that we cannot go into the depth of each article, which means that some nuances may be insufficiently explored in the presentation.

## 4. Results/Findings

### 4.1. The Concept of Participation—The "What" in Child Participation

"Participation" is a complex and multi-dimensioned term that covers a diverse range of practices and settings and is often used without being theoretically explained and operationalized (Percy-Smith 2011; Bessell 2011; Horwath et al. 2012; Tingstad 2019). It can be difficult to arrive at a clear understanding of the concept, partly as there are different terms being used to explained and conceptualize participation. (Healy and Darlington 2009; Križ and Skivenes 2017; Sæbjørnsen and Willumsen 2017).

The terms most frequently used synonymously or in addition to participation are "collaboration", (Husby et al. 2018b), "cooperation" (Vis and Fossum 2015; Thørnblad and Holtan 2012), "consulting" (Mitchell et al. 2010), "involvement" (Goodyer 2016) and "engagement" (Cudjoe et al. 2020; Arbeiter and Toros 2017). "Listening" and "children's voice" are other central terms used in reference to child participation (Nybell 2013; Goodyer 2016; Križ and Roundtree-Swain 2017). When "voice" is used as a metaphor for participation but is not described further, as in the study by Heimer et al. (2017), interpretation of the word is left to the reader. A ticker description, provided as a framework for the analysis (Nybell 2013), clarifies the concept. For Roose et al. (2009), voice is described as participatory work, which is a continuous dialogue where the social worker is the reflexive partner enabling the child to express his/her views. Voice seems to be connected to the child's point of view and may be what Archard and Skivenes (2009) denotes as the child's authentic voice, the child's own view of the matter.

Researchers are proceeding in various ways to elaborate the concepts. Article 12 in the UNCRC seems to form a basis for a definition of "participation" for all the included articles, and expresses understandings of central aspects of participation: children should be adequately informed and be encouraged to express their views, and their opinions should be given due weight in decisions involving them.

Pölkki et al. (2012, p. 108) present a general definition which is not specifically aimed at children's participation: "Generally, participation can be defined as interaction; belonging; integration into and influence on society. It also relates to issues of power and empowerment". Pölkki et al. (2012) apply this general definition to children's participation by relating their understanding of the concept to the UNCRC, defining participation close to the wording of article 12. Others relate the concept of children's participation to the relational process and the power relationship between the child and the social worker (Roose et al. 2009; Thørnblad and Holtan 2012; Nybell 2013; Carter 2014, Aamodt 2015). Nybell emphasizes the importance of context and power relations in the exploration of the child's voice, while Carter (2014) defines participation as the handing over of power to the child by the adult counselor as well as the child "demanding" it from the adult counsellor. Larsen (2011) includes two different forms of participation in her definition, where "formal participation" explores children's involvement in decisions and knowledge about CW, and "everyday participation" measures the influence on activities and situations the child encounters in everyday life.

Not everyone operates with what appear to be clear-cut definitions. Most authors explain their understanding of the concept of participation by referring to models, theories and other studies, where a few contextualize the concept and own study only by referring to other research. To gain insight into which participation practices the studies report from, the context of participation is essential for the clarification of the concept.

*4.2. The Context of Child Participation*

The meaning and expression of child and parent participation are contingent on context (Healy and Darlington 2009). These terms are intended to describe participation practices relevant to CW. There is considerable disparity in the extent to which these studies from 5 continents account for the cultural and local contexts from which they report. Pölkki et al. (2012) frame their study by presenting prerequisites for children's participation in the Finnish context and explain legislation and the social worker's room for action in Finnish CW organizations. Križ and Roundtree-Swain (2017) and Križ and Skivenes (2017) give the reader a similar insight into local conditions in The Netherlands by outlining local laws and organizational conditions that are considered relevant to children's participation, and, therefore, also to their study findings and discussion. Kosher and Ben-Arieh (2019), on the other hand, present a framework for their study by referring to participation models, several international studies of children's participation in CW, and international implementations of children's participation in CW. The local conditions in Israel, however, are not explicitly accounted for. The focus of the study is that social workers' conceptualization of children's participation impacts on how the social workers act, and they try to find explanations for the gap between attitudes and practice. When findings then point to differences associated with the social workers' place of work, and between "regular" and "legal" social workers, the limited contextualization makes it challenging to gain good insight into local contexts, to transfer these findings to other countries and CW organizations, and to compare this study to other studies.

The authors in this review relate children's participation to essentially three different participation roles in CW: the child as a consultant who gives advice to improve services (Mitchell et al. 2010; Woolfson et al. 2010), the child as the center of the decision-making (Vis and Thomas 2009), or as the recipient of measures, where the mutual dialogue is both the tool and the purpose of the meeting (Larsen 2011; Carter 2014). The participation role may also be integrated in parental participation (Healy and Darlington 2009; Arbeiter and

Toros 2017). Inchaurrondo et al. (2018) see children's participation as an element of positive parenting, since it is based on the best interest of the child.

According to Sæbjørnsen and Willumsen (2017), research on children's participation tends to come from three areas of practice: child protection casework and meetings, family group conferences, and reviews and meetings for children in care. Most studies in this review discuss the whole CW process, or they are not specific about which CW phases or tasks they are referring to. Participation in decision-making seems to dominate, whether in the investigation phase or directed at children under care. Few studies focused on children's participation in the intervention phase, regarding decisions about the measures the family received. Attendance and experiences with case conferences dominated. The exceptions are Carter (2014), who studied children receiving community-based interventions for children exposed to violence, and Larsen (2011), who explored children's experiences with their support persons and support families. A couple of the authors explore the conditions of participation in CW practices (Warming 2011; Nybell 2013: Thørnblad and Holtan 2012; Aamodt 2015), and aspects of CW cooperation practices with children (Roose et al. 2009; Gulbrandsen et al. 2012; Ulvik 2015; Husby et al. 2018a, 2018b).

Differences in organizational cultures, understandings of their role and commitment between professionals cause major differences between municipalities, agencies and professionals in children's opportunity to participate (Healy and Darlington 2009; Pölkki et al. 2012; Vis and Fossum 2015; Rap et al. 2018; Kosher and Ben-Arieh 2019). The studies confirm inequalities between different CW practices, but they provide limited information on the contextual conditions of culture, organization, and work processes. As an example, Pölkki et al. (2012) report that children in out-of-home placements experienced more satisfactory participation with CW after moving away from home than in the assessment phase. Heimer et al. (2017) found that children's voices were stronger in the first phase and weak when they were placed out of home. This shows that referring to children's experience of participation in the child welfare case, in general, gives less specific and useful insight than in articles like these, where the authors are specific about the context.

All the studies examined account for age, some also account for gender and, to a lesser extent, they account for whether the children in question are under investigation, receiving measures, or under care. On occasion, the studies say something about the children's behavior or whether they have special needs. However, the studies do not spend a lot of time exploring child characteristics and how they differ from each other; for example, in terms of interests, needs and life situation. This applies both to when the children participate in the research themselves and to when they are the focus of these studies.

The studies report interesting results; however, it is unclear how much of it is transferable. Inequalities between workplaces, teams and work phases can represent different opportunities for participation (Pölkki et al. 2012). It is, therefore, important for researchers who look at child participation to account for the context in which the study is conducted—not only the country, but also which phase of the CW process and which interventions the study refers to.

The concepts used in these studies, and which child protection contexts they examine, are essential information for understanding which participatory practices the studies report on. How do the studies proceed to analyze the CW participation practices?

*4.3. The "How" in Child Welfare—Indicators of Participation*

Based on different participation models, the studies describe indicators to measure the extent, quality and impact of participation. The UNCRC has had a clear and important influence on how countries, systems and researchers think about child participation. All of the studies in this review refer to the UNCRC, and a significant amount of the researchers measure participation by using the words from the UNCRC: "being informed", "being listened to" and the child's opinion, which should be given "due weight".

Several of the authors refer to models as indicators of participation in their studies (Bruce 2014). The most commonly referred to models are Hart (1992) "Ladder of young

people's participation", a metaphor to separate different levels of participation, from manipulation on the lowest level, to the highest level where decisions are made in partnership between child and adult, and the child's initiative. This model is often used as a useful index to examine CW workers perceptions of children's participation, and their statements assessed in line with each step of the ladder (Križ and Skivenes 2017). An example of such a study is that of Cudjoe et al. (2020), which used Hart's model as a theoretical framework to guide the study about children's experiences in child protection meetings: being informed, presenting their view, and having influence on decisions. For those who see Hart's ladder as hierarchical, the expectation of a qualitatively better devotion to participation in each step is criticized because the levels do not necessarily work separately and in chronological order (Roose et al. 2009; Diaz et al. 2018).

Shier's model (Shier 2001), which is described by the author as an additional tool for practitioners, to explore different aspects of participation, describes five levels of participation, from listening to children, to children sharing power and responsibility for decisions on the highest level. In addition, the model includes three stages of professionals' commitments at each level: openings, opportunities and obligations. This model is referred to as an alternative (Pölkki et al. 2012) or additional model (Diaz et al. 2018; Dillon et al. 2016).

Treseder (1997), who presents a modified version of Hart's ladder model (Hart 1992) is another model referred to by several authors (Diaz et al. 2018; Carter 2014; Roose et al. 2009). Treseder moved away from the hierarchical notion of the ladder, and developed a circular model describing five degrees of participation, emphasizing that different types of participation are relevant to different contexts and circumstances.

Theories and models seem to function as inspirations and can be good tools to organize thinking and create structure and focus on the analysis (Sohlberg and Sohlberg 2013). Roose et al. (2009) applied discourse analysis, as described by Fairclough, analyzing CW documents and looking for the children's perspectives in CW reports. Aamodt (2015) explores children and social workers' leeway for children's participation based on observations in CW conversations using Foucault's dispositive analysis. Nybell (2013) looks for both conceptual clarity and an analytical framework in Honneth's theories, while Warming includes Hart's ladder, Bourdieu's theoretization of power dynamics and Delanty's concept of citizenship identity in her analytical frameworks. She explores the intersection between client child's position, state logics in the form of new governance interests in childhood and children's rights. Fylkesnes et al. (2018) are theoretically informed by Fraser, identifying barriers and facilitators in normative and economic power structures influencing the ethnic minority of children's opportunity for participation. Recognition and power are brought in as central concepts in relation to participation, used as a lens to examine and conceptualize participation. Studies in this review refine the concept and convey their perspectives by referring to theories and by reflecting on them to a greater or lesser extent.

Regardless of models, attendance at meetings, knowledge and understanding of CW is the most frequently used indicator of participation (Healy and Darlington 2009; Vis and Thomas 2009; Vis et al. 2012; Pölkki et al. 2012; Bruce 2014). Influence in decision making in formal meetings, such as case conferences, review meetings (Muench et al. 2017; Vis et al. 2012; Križ and Roundtree-Swain 2017; Vis and Thomas 2009; Bruce 2014; Diaz et al. 2018; Cudjoe et al. 2020), or interprofessional teams (Sæbjørnsen and Willumsen 2017), are measured.

Dillon et al. (2016) problematize that child participation seems to default to the child attending meetings. For participation to be effective, children need to understand what is at stake and to be engaged in an ongoing dialogue. "Informed" is often seen as a prerequisite for the "next levels of participation", "heard" and "given due weight", and this is measured by what information the children retain from meetings or decisions made by the CW, and to what degree they understand the reason for the interventions. For Rap et al. (2018), this includes that the child must be aware of the options available, prepared for the particular meeting. Paulsen (2016) and Van Bijleveld et al. (2015) describe being present and having influence as two different forms and levels of participation. They distinguish between

consultative and collaborative participation. According to Paulsen (2016), consultative participation is about the child being present and what the child says is being documented, while collaborative refers to situations where the child's opinions influence the decisions. For Van Bijleveld et al. (2015), collaborative participation means the child's views are taken into consideration.

Different kinds of participatory activities and relationships are appropriate in different settings (Healy and Darlington 2009, Dillon et al. 2016; Carter 2014). In some settings, it is neither possible nor desirable to put the child in control, which is described as the highest position in the hierarchy by Hart; for example, in therapeutic settings such as those that Carter (2014) suggests.

The extent to which the CW has succeeded in implementing children's right to meaningful and effective participation is attempted graded. Meaningful participation is connected to the quality of participation and the children's experiences of feeling listened to and taken seriously (Woolfson et al. 2010; Dillon et al. 2016; Kennan et al. 2019).

What they say should also be heard. Studies try to measure the effect by asking children about their experiences of meetings and contact with social workers; are the children listened to? Archard and Skivenes (2009) problematize the concept of hearing. Hearing children should not be a final goal, but a means of achieving the rights of the children. They provide a clear content beyond seeking children's views and claim that we should not seek the view of the child simply to democratize the child's competence to decide for himself or herself (ibid). Just as important, they might play a conclusive role in helping the adult in decision-making to judge what is in the best interest of the child. The child's point of view should have value in itself as an element in the decision-making process (ibid).

It is difficult to measure what it means for a child to express their views freely. The studies claim that participation provides several types of benefits for children. What do the researchers in this review see as the purpose for children to participate?

### 4.4. Why Should We Cooperate with Children in Child Welfare?

All included articles emphasize children's right to participation as the most central reason why children should participate. In addition, it is argued that children are citizens and service users and share the same fundamental human rights (Pölkki et al. 2012; Larsen 2011; Nybell 2013) and civil rights (Dillon et al. 2016; Paulsen 2016; Muench et al. 2017) as adults.

However, rights are not the only reason why it is argued that children should participate. Inspired by Bessell (2011), we have categorized the author's arguments for children's participation in CW processes into three categories: rights, intrinsic values and instrumental values. Some authors only refer to one of the categories, but most frequently they refer to reasons that fall into two or all three of the categories. Intrinsic values, such as children's well-being, self-esteem, self-efficacy and self-confidence, were highlighted by 23 studies. It is argued that listening to children contributes to their resilience (Van Bijleveld et al. 2014), encourages autonomy and agency (Križ and Roundtree-Swain 2017; Rap et al. 2018) and good health for children in contact with CW and potentially protects them from abuse (Vis et al. 2012; Cossar et al. 2014; Dillon et al. 2016; Muench et al. 2017; Balsells et al. 2017; Heimer et al. 2017; Križ and Skivenes 2017; Sæbjørnsen and Willumsen 2017; Kosher and Ben-Arieh 2019). Participation also gives children practice in taking responsibility for their own choices, which will help children make the right decisions and prepare for their futures as adults (Mitchell et al. 2010; Healy and Darlington 2009; Dillon et al. 2016; Van Bijleveld et al. 2014; Muench et al. 2017).

Interventions seem to be more effective and successful when they are better tailored to the needs and daily realities of the child (Archard and Skivenes 2009). This, of course, benefits the child, but it may also be motivated by the success of the service and could therefore be seen as an argument about both intrinsic and instrumental value. Participation could help the child feel connected and committed to the decisions that are taken (Woolfson

et al. 2010), and at the same time motivate the child to accept the decision or the intervention being made (Van Bijleveld et al. 2014), and in this way give better provision and outcome. Ethical reasons are also mentioned as a reason children should participate, which might both be seen as a question of rights and intrinsic and instrumental value (Nybell 2013; Goodyer 2016; Husby et al. 2018a, 2018b; Inchaurrondo et al. 2018).

*4.5. The Purpose of Participation in Child Welfare*

All of these studies agree that children have the right to express themselves; that their opinions should be heard, and that they should be able to influence the decisions taken. The difference between them lies in the fact that some articles see children's participation rights as more than getting hold of the child's already formed point of view. Children's participation could include both the rights-based concept of participation and a social and procedural concept of participation, such as support for understanding, basis for decisions, and exercise in participation (Winter 2010; Gulbrandsen et al. 2012; Seim and Slettebø 2017).

The relational aspect is considered as a prerequisite for participation to be meaningful to the child. Although participation should be in the form of a process and not a one-off event (Vis et al. 2012), there seems to be a difference in how these authors think about what the central goals of participation is. According to Ulvik (2015), the dominant term "effective participation" in the research literature on children's rights analytically privileges the result over the process, since many authors define participation by its result. A different perspective, in addition to being rights-oriented, is offered from those studies more concerned about participation as a relational concept, and as a part of a relational discourse (Roose et al. 2009; Nybell 2013; Aamodt 2015; Seim and Slettebø 2017; Husby et al. 2018a, 2018b). Participation always takes place in a social context. The professional can help the child make sense of what is happening and the challenges they face. Gulbrandsen et al. (2012) argue that child participation is about more than achieving a higher quality of process and argue that we need a broader participation concept than the legal one offered by the UNCRC. This concept may include, but should not be limited to, decision-making. Seeing the child's point of view as something that can be formed during the process, rather than something fixed to be extracted by social workers, naturally requires a more process-oriented understanding of child participation. This collaboration with the child seems to go beyond what is meaningful to the child. Children's views, meanings and opinions are not completed tasks, as both children and their conditions are subject to change. Children have the right and may need to be assisted in forming views and articulating experiences (Stang 2007; Bruce 2014; Jobe and Gorin 2013; Cossar et al. 2014; Van Bijleveld et al. 2014; Carter 2014; Križ and Skivenes 2017).

Negotiation through dialogue requires that children know the subject and consequences for both the dialogue and the decisions being made. The topics that children are encouraged to talk about in CW matters are not easy to articulate. The decisions to be made often involve past traumatic events and difficult choices for the future. Such understandings are not created in a social vacuum but are developed in interaction with other people (Gulbrandsen et al. 2012). This understanding of participation is neither hierarchical in its view of what has been achieved or concerned about the result and the effect (ibid.).

The discourse of rights dominates the studies. In addition, some explore the quality of the relationship between child and social worker, including aspects of power. A few explore the scope of action for children's participation in CW. Citizenship and human rights are mentioned by a few. Warming (2011) explores more in-depth children's participation and citizenship in the context of the intersecting logics of the field of social work with children. She points to the structural forces, such as new childhood governance policies and children as the materials for the future, combined with new public management and the focus on effectiveness and outcome shape client children's rights.

## 5. Discussion

What is included in the concept of child participation has consequences for the understanding of the purpose of child participation, which in turn has a decisive effect on how we proceed to explore the CW service's participation practices.

The idea of participation generally is related to the idea of the liberty and freedom of humankind, and several authors emphasize the importance of being heard and the ability to be represented (Honneth 1995; Fraser 2000). This has also strongly influenced the discourse on children's place in society, of which the UN Declaration on the Rights of the Child is an example.

Children's participation is a universal right, embodied in various practices, dependent and relationally linked to their cultural environment. Critical voices claim that children's rights are presented in practice and policy as the new "norm" that children need to adapt to (Reynaert et al. 2009; Mason and Bolzan 2010; Quennerstedt 2010; Quennerstedt et al. 2018). The question is whether a universal norm of participation is relevant for all children, and in all cultural and social contexts in which children operate. CW work processes have different tasks, goals and procedures for the work. Without the problematization of the different cultural environments and the contexts in which the CW services invites the children to participate in, the concepts, goals and tasks will remain unclear. Children's expressions are created in social and cultural contexts, with a specific purpose, and often with a significant degree of ambiguity (Spyrou 2011; Tingstad 2019).

When the authors do not sufficiently explain the context from which they report, they give the impression that "children's participation" always implies the same thing, regardless of context, which leaves interesting differences undetected. This consensus thinking related to children's right to participate may lead the discussion to a technical debate on the most effective and efficient way to implement children's rights, and how best to monitor and organize this implementation (Reynaert et al. 2009). We risk losing the content, the purpose of the participation, and what benefit it has for different children.

In this review, we found that references to models, indicators and principles that rank the degree of participation are linked to concepts such as "voice" and "being listened to". The term "voice" seems to be a universal response to children's right to participation. Tingstad (2019) warns against uncritical and simplified use of voice as a term. She asks if voice is something to be extracted, as if voice is something one has, ready formulated? Or is voice something that is created in a dynamic interhuman communication that is both complex and context-dependent? (Gulbrandsen et al. 2012; Ulvik 2015; Seim and Slettebø 2017; Tingstad 2019). Children in contact with CW navigate in a variety of settings where both diverse relationships and time available to listen challenge how social workers intervene in the power laden settings (Nybell 2013).

Models and theories are presented as starting points, inspiration or indicators for measuring how CW meets the children's right to participation. If the goal is to move on from repeating measurements of children's lack of participation in CW, evaluations of impact, having a say, and using models and theories may not be the best solution. According to Malone and Hartung (2010), Hart himself criticized the model he developed for cultural bias as currently misused as a comprehensive tool for understanding and evaluating children's participation in projects (ibid.). The prevalence of superficial references to models could possibly be attributed to the lack of theoretical clarity that we also find in this review. Instead, participation is held up conceptually through "a scaffold of ladders, degrees, levels, enabling environments and supporting adjectives, as meaningful and ethical" (Theis 2010, s. 344).

The varying degree of account of one's own theoretical and conceptual position provides a basis for different understandings of the research presented. Taking for granted that we all understand a concept in same way does not remove the concept of participation as ambiguous and value laden (Bal 2002), and we risk underestimating the fact that the terms we use are often potentially controversial concepts that communicate our view of children and their position in the world (Tingstad 2019). When the studies, to a limited

extent, succeed in conceptualizing the term, they provide inadequate contextualization of what they are talking about, or fail to include children's participation as part of the research question. When it is a central theme in the results, this has consequences for further research. It becomes unclear whether we have an intersubjective understanding of concepts and findings, potentially leading to consequential errors in the field when researchers reference each other as support for understanding their own findings.

A prominent discrepancy between the studies in this review are whether they promote a rights-based discourse, or whether they emphasize that participation is a process that develops through social interaction over time. Most studies in the review are interested in the result of participation, while only a few emphasized the process. Although many see being a part of a participation as an important prerequisite for a good result, others point to the recognition that a child's perspective should not simply involve hearing the child's ideas or information the child that they can contribute. Rather, the purpose of the participation process involves understanding the importance of the child's experience in the context of the child's everyday life.

Warming (2011) and Theis (2010) contribute a broader concept and theoretical conceptualization of child participation, which is in line with what Percy-Smith and Thomas (2010) called for 10 years ago. The UNCRC has been an important driving force for implementing child participation, but they claim that time has come to look beyond Article 12, and even beyond the UNCRC, to other interpretations of children's rights. They suggest real citizenship through full democratic rights at all levels, and safeguarding children's rights, equality and justice through actively participating in everyday life (ibid.). If we are to get past today's barriers to children's participation, they argue, we must go beyond thinking of children's participation as something different from everyone else's participation. It must become regular practice for everyone work with children in everyday life and in all arenas; at home, at school and for leisure, not just within special events or one-off projects (Liebel and Saadi 2010; Theis 2010).

Listening to children's voices today is often included in the discourses on children's rights (Tingstad 2019). Participation that implies shaping the services adults have provided offers a narrow definition of participation (Percy-Smith and Thomas 2010; Quennerstedt 2010). A reorienting of children's participation goes beyond exercising the right to "have a say" or being "listened to". A broader concept of active citizenship for children consists of more than speaking and being heard.

The starting point for many studies concerning children's participation is the lack of children's opportunity to "have a say" in decision making, where the question is what the purpose behind these calls for participation is. (Mannion 2010). The various studies' claims of positive outcomes and a better future for children as the results of participation are usually not concretized or questioned. "Children" covers a very diverse group. What is appropriate for some may not suit another, whatever context and whatever work is to be done. Participation is claimed to provide training and socialization. When is participation training for making democratic decisions, and when does it simply help legitimize adult perspectives (Nybell 2013)? Another question is whether socialization to become a better citizen in the future challenges the perspective of the child as a rights bearer and citizen here and now. (Kjørholt 2002). The socialization paradigm, which emphasizes children's development towards becoming mature human beings in the future may, in some ways, be seen as contradictory to the construction of the child as a rights-holder in modernity, stressing children's rights as citizens in the here and now (Warming 2011).

Participation as an integral part of social living, and citizenship seems to be an ontological position taken in Western countries, in which children are prepared for the future and adulthood as full citizens (Wyness 2018). The understanding of children in modern Western countries is linked to value concepts, such as liberty, human rights, respect, and equality (Kjørholt 2004). These values correspond to positions that promote equality and dignity in egalitarian societies and are based on liberal values, such as the importance of independent judgment, and the possibility of differences in values and preferences. The

individualistic orientation that values unconditioned personal freedom is a core value in Nordic countries and is central in other Western countries (Kjørholt 2002). The same values do not have the same position worldwide. There may seem to be a surface consensus on the content of the term. The wording is that children should participate, but the practice is different (Tingstad 2019; Reynaert et al. 2009).

How we understand participation influences children's ability to participate. When participation is communicated and forms the basis for participation practices without further questioning and redefinition, we can get different results than intended. The "norm" (Reynaert et al. 2009; Quennerstedt et al. 2018) to always involve children, which Arbeiter and Toros (2017) refers to as the "current child protection philosophy" might end up in what Aamodt (2015, s.80) warns against: "The child's right thus becomes the child's duty where one can think that the duty is to ensure the child's right. It thus emerges a right the child has a duty to follow".

The review indicates that participation may imply different things, and the authors include varying degrees of explanation of how they understand the term. The contexts are different in terms of geography, culture, and work phases, and for this reason, also the aim of the CW and the child's prerequisites to participate. The studies refer to, build on, or expand upon reasoning between the authors, which builds a discourse in the field. A critical question is what analytical power the term has in the span of different contexts and understandings. When the knowledge base is unclear, it can contribute to more polarization, rather than complementing and further exploring different child participation processes (Tingstad 2019). While most researchers are concerned with exploring barriers to the implementation of children's rights in CW, others express concern about children's right to participation becoming a "technical discourse," which can lead to overgeneralization and ritualization of practice (Reynaert et al. 2009; Ulvik 2015). The result is an understanding of children's rights that is measured by what the children verbally express. If this is not problematized in research, politics and practice, the practice is left to techniques and simple solutions where the attainment of rights and responsibilities is proportional to what adults perceive (Reynaert et al. 2009).

The Western tradition of seeing participation as the individual right of to speak and be heard in decision making, offers a narrow understanding of the concept. Participation could be more than a specific type of communication with children, arranged for specific purposes. A broader participation concept includes active and routine inclusion in vital social processes, situating children in society with the opportunity to practice and develop participation from an early age in their communities (Liebel and Saadi 2010; Percy-Smith and Thomas 2010; Theis 2010).

Implications: Many of the studies in our review refer to the child welfare workers' understanding of the concept of child participation as an explanation for the lack of participation. Our findings show the need for researchers to also clearly explain their understanding of the concept of participation. In this way researchers can contribute to the diversity and similarities of the studies being seen in relation to each other, which might make it easier for practitioners to acquire knowledge. We also see a need to develop a model for children's participation that includes a time variable that captures that children's views are fluxurating and clarifies the process as a goal, thereby limiting the one-sided measurement of participation as children's presence and verbal expression in single events.

For further research, we see that it may be relevant to make a comparison between countries, their characteristics according to national culture, how child welfare services are organized and the perception of the child. It is particularly interesting to compare the different typologies of child welfare systems that include both low-, middle- and high-income countries.

**Author Contributions:** Conceptualization, B.S., A.S.S. and E.M.; methodology, B.S., A.S.S. and E.M.; software, B.S.; validation, B.S., A.S.S.; formal analysis, B.S. and A.S.S.; investigation, B.S.; resources, B.S. and A.S.S.; data curation, B.S.; writing—original draft preparation, B.S. and A.S.S.; writing—review and editing, B.S. and A.S.S.; supervision, E.M.; project administration, B.S. All authors have read and agreed to the published version of the manuscript.

**Funding:** This research was funded by Norwegian Research Council and Trondheim kommune, grant number 272849.

**Institutional Review Board Statement:** Not applicable.

**Informed Consent Statement:** Not applicable.

**Data Availability Statement:** Not applicable.

**Conflicts of Interest:** The authors declare no conflict of interest.

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
