# Peer review of "The What, Why and How of Child Participation—A Review of the Conceptualization of “Child Participation” in Child Welfare"

_socsci, doi:10.3390/socsci10020054_

Round 1

Reviewer 1 Report

The what, why, and how of child participation

This paper includes a scoping review of 46 peer reviewed articles to sillustrate how various authors have conceptualized the concept of child participation in child welfare.  Given the increasing interest in this concept in the field, the paper is a welcome addition.

Introduction

The authors do a good job of framing the context for the reader.

Methods

The authors do a modest job of describing their methods and the limitations.  The authors indicate that they included studies that relied upon primary data and also studies where the general concept of participation was included.  It would be helpful if the authors could please indicate how many of their total articles (n=46) were based on primary data, and how many were conceptual articles.  Further, more information on the total number of articles harvested in their initial data collection process would be helpful.  The authors indicate that “86 articles remained.”  Does that mean that they originally collected 106 articles?  Please describe more about what was excluded to reduce the sample to 46. 

Can the authors please say more about their three primary measures?  As I understand, they were looking for three things: (1) how they explain and contextualize the concept of participation; (2) how they operationalize the concept; and (3) “what they express about why children should participate the purpose of participation” (note that there’s likely a typo in this last section of the authors’ sentence).  I believe domain #1 is actually two domains: (1a) How is child participation defined? and (1b) What is the context that frames a definition of child participation? And domain #3 is actually two domains: (3a) Why is participation perceived as beneficial; and (3b) What is the goal of participation? 

Can the authors clarify if all three domains were required in order to be included in their review?  If an article could only include one or two of the three domains, how many of these articles were included?  A table showing how many articles were included (46), and how many of these included all three domains, only two (which two), and only one (which one) would be helpful.

Can the authors please clarify if the whole article was reviewed for all of the 46 articles in their sample, or just the title and abstract?  If the process was not systematic across all of the articles, please clarify why and clarify how many articles received a full review?

The various contexts of child welfare systems

This section should be moved so that it precedes the Methods.  The reader needs to understand the child welfare context before moving into the study itself.  In this section, the authors seem to indicate that they measured an additional domain: Did the paper define the study’s context as it relates to culture.  If that was indeed a measure in the study, the authors must include this in the Methods section and operationalize what they mean by “defining the study’s context as it relates to culture.” Further, the authors seem to indicate that they collected data on the age of the child that was being discussed, and other child characteristics.  If these were data points, they need to be included and described in the Methods section.

This section also indicates that service-oriented systems (as in the Nordic/ Scandinavian countries) have a child-centered approach, where children’s rights are embedded in the frame.  If so, did the authors systematically analyze the articles that included service-oriented systems to determine if their frame for describing children’s participation was different from the other countries?  Finally, it would seem that the authors have, in part, answered their own question in this section as they describe the service-oriented systems as countries where: “Children are given the opportunity to express their views and opinions and are involved in matters concerning them.” Are these not indicators of “child participation?” If not, the authors need to clarify for the reader what they were looking for in the data that either matched this definition, or deviated from it.

Results

I’m afraid I don’t understand the paragraph that begins: (p. 6) “The authors in this review relate…”   Are these findings they are reporting, or are these interpretations of findings?

The authors sometimes offer findings from other studies, rather than definitions.  For example, at the bottom of p. 6, why do the authors tell the reader about the findings on children’s participation from the Polkki et al, study? In this paragraph, they tell the reader about children’s experience of “participation,” but the purpose of this paper is to define “participation.”

  1. 8, line 381. I’m afraid the sentence doesn’t make sense. Perhaps there are words missing?

“Why should we cooperate with children in child welfare?“  In the methods, the authors use the term “participate;” here, however, they use the term “cooperate.”  I see these concepts as very different.  Can the authors clarify why the sub-title includes the word “cooperate”?

Discussion

The discussion offers a good overview of the findings.  I was hoping the authors would take their findings and offer an approach for the field to help sharpen and clarify language and intent.  In other words, the article leaves the reader with a good sense of how others have defined “participation” in the past, but does not give much of a roadmap for the future.

Author Response

Open Review

Review report 1

Comments and Suggestions for Authors

The what, why, and how of child participation

This paper includes a scoping review of 46 peer reviewed articles to illustrate how various authors have conceptualized the concept of child participation in child welfare. Given the increasing interest in this concept in the field, the paper is a welcome addition.

Introduction

The authors do a good job of framing the context for the reader.

Methods

The authors do a modest job of describing their methods and the limitations. The authors indicate that they included studies that relied upon primary data and also studies where the general concept of participation was included. It would be helpful if the authors could pleas

  • Indicate how many of their total articles (n=46) were based on primary data, and how many were conceptual articles.

40 articles where reporting from primary studies, while 4 were based on primary studies, but data were to a lesser extent reported from and served as illustrations and examples for their theoretical exploration of the concept of 2 articles. The articles from Gulbrandsen, Seim & Ulvik (2012) and Seim & Slettebø (2017) where based on data from their primary projects and Warming (2011) and Ulvik (2015) used examples from their primary studies (*)

2.- more information on the total number of articles harvested in their initial data collection process would be helpful. The authors indicate that “86 articles remained.” Does that mean that they originally collected 106 articles? 

3.-Please describe more about what was excluded to reduce the sample to 46.

N= 1496 from all databases. Titles were read and duplicates, articles from other fields than child welfare   was excluded

N=318 remained after that, and then titles and abstracts were examined. Articles focusing child participation in the justice system, referral phase, participating with parents,

N=86 articles were identified as potentially relevant and the full text was examined. Examples of articles excluded were studies based on literature studies, studies focusing communication skills and developing tools in communicating with children or use of language interpreter in conversations with children.

40 articles were in line with the inclusion criteria

4 articles were identified in in reference lists. They did not report from primary projects to the same extend as the other 40, but have a more theoretical argument related to exploring the concept of child participation ( *)

  1. Can the authors please say more about their three primary measures? As I understand, they were looking for three things: (1) how they explain and contextualize the concept of participation; (2) how they operationalize the concept; and (3) “what they express about the purpose of participation” (note that there’s likely a typo in this last section of the authors’ sentence). I believe domain #1 is actually two domains: (1a) How is child participation defined? and (1b) What is the context that frames a definition of child participation? And domain #3 is actually two domains: (3a) Why is participation perceived as beneficial; and (3b) What is the goal of participation?

This part is reformulated, and hopefully clarified. Please let us know if there is still a need for more information.

5.Can the authors clarify if all three domains were required in order to be included in their review? If an article could only include one or two of the three domains, how many of these articles were included? A table showing how many articles were included (46), and how many of these included all three domains, only two (which two), and only one (which one) would be helpful.

These 3 domains were a result of the analysis process, and not an inclusion criterion. Nearly all articles included all three domains, to different degrees. If you mean that there is still a need for a table, please let us know.

  1. Can the authors please clarify if the whole article was reviewed for all of the 46 articles in their sample, or just the title and abstract? If the process was not systematic across all of the articles, please clarify why and clarify how many articles received a full review?

44 articles received a full review

The various contexts of child welfare systems

  1. This section should be moved so that it precedes the Methods. The reader needs to understand the child welfare context before moving into the study itself. In this section, the authors seem to indicate that they measured an additional domain: changed in line with the reviewer's comments
  2. Did the paper define the study’s context as it relates to culture. If that was indeed a measure in the study, the authors must include this in the Methods section and operationalize what they mean by “defining the study’s context as it relates to culture”. Clarified what is meant by “culture”.
  3. Further, the authors seem to indicate that they collected data on the age of the child that was being discussed, and other child characteristics. If these were data points, they need to be included and described in the Methods section. Included and described in methods.

  1. This section also indicates that service-oriented systems (as in the Nordic/ Scandinavian countries) have a child-centered approach, where children’s rights are embedded in the frame. If so, did the authors systematically analyze the articles that included service-oriented systems to determine if their frame for describing children’s participation was different from the other countries? Unfortunately, this was not a part of the analysis, but is described as a need for further research.
  2. Finally, it would seem that the authors have, in part, answered their own question in this section as they describe the service-oriented systems as countries where: “Children are given the opportunity to express their views and opinions and are involved in matters concerning them.” Are these not indicators of “child participation?” If not, the authors need to clarify for the reader what they were looking for in the data that either matched this definition or deviated from it. Clarified that this is an intention of these systems, and not necessarily the practice.

Results

  1. I’m afraid I don’t understand the paragraph that begins: (p. 6) “The authors in this review relate…” Are these findings they are reporting, or are these interpretations of findings? Clarified, please let us know if it is still unclear
  2. The authors sometimes offer findings from other studies, rather than definitions. For example, at the bottom of p. 6, why do the authors tell the reader about the findings on children’s participation from the Polkki et al, study? In this paragraph, they tell the reader about children’s experience of “participation,” but the purpose of this paper is to define “participation.”

We have tried to change that section to make it clear that comparison of findings presupposes good insight into the context in which studies and findings have been carried out. We have clarified that these references to article findings are included as examples, and clarified the significance of this contextualization for the understanding Please let us know if there is a need for further clarification

 14.12.-8, line 381. I’m afraid the sentence doesn’t make sense. Perhaps there are words missing?

"Point to" has been changed to “suggest”s. Please let us know if it is still unclear

 15.“Why should we cooperate with children in child welfare?   In the methods, the authors use the term “participate;” here, however, they use the term “cooperate.”  I see these concepts as very different.  Can the authors clarify why the sub-title includes the word “cooperate”? Changed in accordance with reviewers' comments

Discussion

16.The discussion offers a good overview of the findings. I was hoping the authors would take their findings and offer an approach for the field to help sharpen and clarify language and intent. In other words, the article leaves the reader with a good sense of how others have defined “participation” in the past but does not give much of a roadmap for the future.

The last section of the discussion has been changed in line with the reviewer's response

Reviewer 2 Report

The analysis performed by the authors is extremely interesting, in the context of the changes that occur in modern society. Parents and society are changing.

The authors use electronic databases to find studies and articles. The authors indicate that after the exclusions they keep 44 articles for analysis and state that their study has potential limitations.

I recommend the authors to develop the description of the search strategy, showing even through concrete examples how they queried the databases. I recommend the authors to clarify the terms used (for example: N = 20). It is not sufficiently explained why the authors did not go into the depth of the selected articles, the large number not being an adequate justification. It would be useful to know if 44 or 46 articles were used for analysis.

In the chapter "Results / Findings" the authors make a detailed presentation of the information collected. As the authors mention in Chapter 3 that they selected studies from 13 countries, I recommend that the authors include a correlation of the information collected with the country. Country characteristics are relevant to the study. Are there no differences in mentality between different countries?

Author Response

Reviewer report 2

The analysis performed by the authors is extremely interesting, in the context of the changes that occur in modern society. Parents and society are changing.

The authors use electronic databases to find studies and articles. The authors indicate that after the exclusions they keep 44 articles for analysis and state that their study has potential limitations.

  1. I recommend the authors to develop the description of the search strategy, showing even through concrete examples how they queried the databases. I recommend the authors to clarify the terms used (for example: N = 20). It is not sufficiently explained why the authors did not go into the depth of the selected articles, the large number not being an adequate justification. It would be useful to know if 44 or 46 articles were used for analysis.

We have used 44 articles for analysis, we’re sorry but there has been the wrong number presented. This is now changed to the right number (44). We did go into the depth of all 44 articles in the analysis but did not present all 44 articles in depth due to the large number. This is now clarified in the text.

  1. In the chapter "Results / Findings" the authors make a detailed presentation of the information collected. As the authors mention in Chapter 3 that they selected studies from 13 countries, I recommend that the authors include a correlation of the information collected with the country. Country characteristics are relevant to the study. Are there no differences in mentality between different countries?

We have included information in the methods section about how we have used this information (about country) in the analysis. Differences in mentality between different countries is unfortunately not a apart of this analysis, but we have emphasized this as a need for further research.   

Round 2

Reviewer 1 Report

The authors appear to have addressed my concerns.  I appreciate their efforts.  The paper now reads more clearly.